# The latitudinal speciation gradient in freshwater fishes: Higher speciation across assemblages at higher latitudes in the northern hemisphere

Juliana Herrera-Pérez[1]*, Juan Carvajal-Quintero[2], Axel Arango[3],
Daniel Valencia-Rodríguez[1], Ana Berenice García-Andrade[4], Pablo Tedesco[5],
Fabricio Villalobos[1]*

**1** Laboratorio de Macroecología Evolutiva, Red de Biología Evolutiva, Instituto de Ecología A.C., Veracruz, México, **2** Department of Biology, Dalhousie University, Halifax, Nova Scotia, Canada, **3** Center of Computational and Theoretical Biology (CCTB), Würzburg University, Würzburg, Germany, **4** Center for Advanced Systems Understanding (CASUS), Helmholtz-Zentrum Dresden-Rossendorf (HZDR), Dresden, Germany, **5** Université de Toulouse, Toulouse INP, CNRS, IRD, CRBE, Toulouse, France

\* juliana.herrera.p@gmail.com (JH-P); fabricio.villalobos@gmail.com (FV)

## Abstract

Speciation rates are a key driver of diversity patterns and are often used to explain the latitudinal diversity gradient (LDG). However, latitudinal variation in speciation rates at both assemblage and species levels remains poorly explored in freshwater fishes. This highlights a gap in understanding the mechanisms driving geographic biodiversity gradients in freshwater fishes. Here, we investigated the latitudinal speciation gradient in freshwater fishes, using a comprehensive database of freshwater fish distributions and phylogenetic relationships of Actinopterygian fishes at the global scale. We estimated speciation rates using three metrics (BAMM, DR, and ClaDS) and evaluated the latitudinal speciation gradient through spatial and phylogenetic regressions at assemblage and species levels. Finally, we analyzed those patterns based on the species assemblage's phylogenetic diversity and structure. Our results show that areas and species with the highest speciation rates were in the tropics. However, the general assemblage pattern revealed a positive relationship between absolute latitude and speciation rates. This relationship is generally absent in tropical regions below 24.39° and became significant only at higher latitudes, particularly in the Northern Hemisphere. We did not find a significant relationship at the species level, mainly due to the strong influence of hyper-diverse groups like Cichliformes. Species-level findings showed the contribution of particular lineages to the speciation gradient as a whole, while the assemblage-level results emphasize the high speciation rates across the Northern Hemisphere, especially North America, potentially resulting from environmental filtering and dispersal events consistent with glaciation dynamics in the Pleistocene.

**Data availability statement:** All data used in this article has been obtained from the cited references. Processed data and code to repeat the analyses can be accessed from Figshare: https://doi.org/10.6084/m9.figshare.27922032.

**Funding:** Consejo Nacional de Humanidades, Ciencias y Tecnologías (CONAHCYT, México) (doctoral scholarship #1149672). German Centre for Integrative Biodiversity Research (iDiv) Halle-Jena-Leipzig (German Research Foundation FZT-118 grant 202548816) via sDiv and FLEXPOOL. The funders had no role in study design, data collection and analysis, decision to publish, or preparation of the manuscript.

**Competing interests:** The authors have declared that no competing interests exist.

## Introduction

One of the most striking biodiversity patterns is the remarkable number of species found in tropical regions, contrasting with the lower species richness in temperate regions [1]. This well-documented latitudinal diversity gradient (LDG) has been consistently observed across taxa, ranging from microorganisms [2] and plants [3] to vertebrates, including marine and freshwater fishes [4,5], amphibians, birds, and mammals [6]. Multiple hypotheses have been posited to explain this gradient, which can be categorized into three main groups: ecological hypotheses, implying factors related to species coexistence and persistence; historical hypotheses, considering the timing of colonization and the duration and extent of tropical environments; and evolutionary hypotheses, which consider the diversification dynamics of studied clades [7].

Diversification rates defined as the balance between species origination and extinction rates, should play a pivotal role in shaping the LDG. Indeed, a higher diversification in the tropical region compared to the temperate region would be sufficient to generate the current LDG [7]. While higher diversification rates in the tropics can explain their exceptional diversity, whether diversification rates in the tropics genuinely surpass those outside this region as well as their consequences for biodiversity gradients, remain relatively less explored compared to ecological and environmental factors [8]. This is mainly because estimating diversification rates is challenging given the time scale at which they occur and the lack of reliable information such as fossils [9]. To overcome these limitations, current methods infer diversification patterns using reconstructed phylogenies, depicting the evolutionary relationships among extant species [10]. For example, different metrics with different assumptions have been proposed to estimate tip-level diversification rates, which can be interpreted as estimates of present-day rates of speciation or extinction of individual lineages/branches conditional on their past evolutionary history, but provide more reliable estimation of speciation rates instead of net diversification [11,12].

Despite methodological differences, speciation rate metrics have been useful for assessing the role of speciation in shaping the LDG [5,13,14]. Using these metrics, several studies have found that speciation rates are not the primary driver of the LDG. Instead, other factors such as climatic stability and the timing of colonization, have emerged as the key explanatory variables of the LDG [4,13,15]. Other studies have focused on describing the latitudinal speciation gradient with contrasting results across different scales and taxa [8,16]. Varying from positive [5] to negative [17] or no relationship [18] Moreover, the majority of studies exploring the latitudinal variation of speciation have been conducted using taxa distributed in open habitats (such as marine and terrestrial ecosystems) or using assemblages as units without considering species-level patterns [4]. As such, the question of how speciation rates vary across latitude at assemblage and species levels in clades that occupy fragmented habitats that limit spatial connectivity, such as rivers, remains open.

Freshwater habitats, which cover only 0.8% of the Earth's surface [19], are characterized by their confinement within terrestrial landscapes forming units delimited by

drainage basin boundaries. Inside these basins, the landscape is characterized by drainages that describe an arborescent bifurcation of a mainstem and branches that decrease in size and increase in number as one proceeds up the network [20] and by isolated or connected water surfaces or lakes. The configuration of freshwater habitats limits the dispersion routes available to strictly aquatic organisms to the river branches of the drainage network or maintains organisms isolated in the lake system. Consequently, the freshwater configuration and spatial connectivity of these habitats profoundly affect the mechanism leading to speciation, extinction, and dispersal [21]. Indeed, freshwater habitats are more susceptible to the effects of tectonic movements compared to the marine habitat, with land's topography changes altering river courses and species geographic distributions [22]. Such tectonic movements have been linked to the diversification of freshwater taxa, like barrier-displacement allowing dispersal by merging adjacent geographic areas, or barrier-formation resulting in vicariant speciation and heightened extinction risk [21].

Actinopterygian fishes comprise ~35,000 species [23], nearly 50% being freshwater. The similar species richness between marine and freshwater fishes, despite the ample difference in habitable area between both habitats, has incited longstanding interest in understanding the processes driving such pattern. The fragmented nature of freshwater habitats has been suggested as the principal driver of fish species richness through allopatric speciation promoting faster diversification rates compared to those of marine fishes [24]. However, this idea has been debated as freshwater fish richness appears to be primarily driven by exceptional diversification in confined systems such as lakes [14] and in particular clades [25]. For example, in Cichlidae, radiation via sympatric speciation have resulted in high diversification rates and ample species richness [26]. The study of speciation patterns at global scale in Actinopterygian fishes has been facilitated by the development of large calibrated molecular phylogenies [5,27]. For marine fishes, a recent study using range maps and a gridded global domain revealed an inverse latitudinal speciation gradient relative to species richness, with higher speciation rates at higher latitudes, a pattern consistent between assemblage and species levels [5]. Similar findings were obtained by Miller and Román-Palacios [4] for freshwater fishes when evaluating their latitudinal species richness pattern, with higher species richness in tropical basins associated to low speciation rates and low species richness in basins with high speciation rates. However, the study by Miller and Román-Palacios [4] focused on species richness explanations at the assemblage level using species lists within discrete basins to define their geographic ranges [28] but did not consider species-level patterns nor a continuous gridded domain to define assemblages as done for marine fishes [5]. Species-level analysis allows identifying lineage-specific patterns by considering individual species as observational units and describing trait (here, speciation rates) variation across the clade. Complementarily, assemblage-level analysis allows describing the entire geographic structure of mean trait variation by considering assemblages of species across space [29]. Thus, combining both types of approaches helps untangling clade-specific from regional-specific patterns [29].

Here, we evaluated the global latitudinal gradient of speciation rate in freshwater fishes and discussed its association with their latitudinal diversity gradient. We used a comprehensive database of freshwater fish range maps [15] and the recent phylogeny of Actinopterygian fishes [5]. To provide different perspectives on the same pattern and a more robust evaluation, we employed both assemblage-level and species-level analysis [29,30] of the geographical variation in speciation rates, which can help infer its role in explaining the observed LDG of freshwater fish species richness [5,8]. Given the pervasive nature of the hierarchical and bifurcating architecture within dendritic networks across all latitudes and the comparatively limited geographical area at southern latitudes, we anticipated a less pronounced pattern in the latitudinal speciation gradient of freshwater fishes compared to the gradient exhibited by marine fishes. In addition, we evaluated the observed speciation gradient based on the phylogenetic diversity and structure patterns of freshwater fish assemblages across the globe. These patterns can provide insights into the maintenance of biodiversity gradients by describing the roles of the evolutionary process of speciation, extinction and dispersal, which are the ultimate causes underlying such gradients [8,31]. Recent studies have evaluated these phylogenetic structure patterns in freshwater fish but focusing on specific regions independently (e.g., North America [32]; South America [33]), thus a global description is still lacking.

## Materials and methods

### Geographic and phylogenetic data

Geographical distributions were obtained from García-Andrade et al. [15] who reconstructed the geographic ranges (i.e., extents of occurrence) of 12,557 freshwater fish species by creating a convex hull for each species based on its occurrence records and overlapping it with the Hydrobasin layer level 8 (see details in [15]). In contrast to basin-level species lists (as in [4]), using species-specific geographic ranges allows capturing detailed spatial variation of species composition even within single basins, where contrasting assemblages can be found from the upper to the lower parts of the basin (e.g., [34]). In fact, using such range maps to evaluate biodiversity gradients at large spatial scales is the standard macroecological practice [35], which recently started to be more widely applied to study such gradients for both marine [5] and freshwater fishes [15]. We projected our range maps to the Behrmann equal-area projection. We considered freshwater fishes as those species recorded in either freshwater, brackish or both habitats. Species recorded in freshwater and marine habitats were discarded to avoid including anadromous/catadromous or incidental species in our analyses. This classification was based on a taxonomic review using the FishBase database, accessed via the rfishbase 4.0 package [36] on January 3, 2023.

Phylogenetic relationships were obtained from a recent species-level phylogeny of Actinopterygian fishes, both freshwater and marine, comprising 31,516 species of ray-finned fishes [5] from 34,721 known species in 2024 [23]. This phylogeny was built using a 27-gene multilocus alignment of 11,638 species with genetic data (including 6,407 freshwater species) under a maximum likelihood framework and time-calibrated using fossil data. Species with no genetic data were imputed into this molecular phylogeny based on their taxonomic placement using a stochastic polytomy resolution under constant-rate birth-death process. This imputation was repeated to generate a distribution of 100 all-taxon assembled (ATA) phylogenies comprising 31,516 species of ray-finned fishes. The workflow for data acquisition and analyses were conducted according to the flowchart depicted in Fig 1.

### Speciation rates

The inference of speciation rates is method dependent, requiring applying several methods to evaluate pattern robustness [12]. We considered three such methods, each with different assumptions, to derive tip-level speciation rate metrics that estimate this rate at the 'tips' of the phylogeny and thus describe current speciation rates that can be related to the maintenance of currently observed species richness gradients [8]. First, we calculated the λDR statistic [37], using the harmonic mean of this metric for each species across the 100 ATA phylogenies with 31,516 species. This DR metric is computationally efficient and estimates recent speciation rates based on branch lengths and splitting events without relying on a parameterized model [11], is computationally efficient but sensible to incomplete or biased taxon sampling producing high variance [38]. Second, we used the model-based estimates of BAMM [39], including (i) time-varying rate regimes BAMM (λTV) and (ii) time constant rates (λTC), previously estimated by [5] using only the molecular phylogeny (11,638 tips). The BAMM method uses a reversible-jump Markov chain model to infer the number and location of diversification-rate shifts across branches. Although the usefulness of BAMM has been debated [12,40,41], it continues to be widely applied. For example, BAMM-derived speciation rate values are already available for all recognized fish species in the Fish Tree of Life website (https://fishtreeoflife.org/), where we obtained the values for our considered species. Third, we calculated tip-level speciation rates using the ClaDS model (λClaDS), a Bayesian approach where diversification rates change only at speciation events [42]. We applied the ClaDS model to the molecular phylogeny containing only species with genetic data (11,638 tips) using the recent ClaDS2 model available in jPANDA package in Julia [42], considering the same sampling fraction reported in Rabosky et al [5] (Fig 1B). Each of these speciation rate metrics have important differences, with BAMM identifying discrete shifts in diversification regimes (i.e., cohorts) that tend to reduce rate heterogeneity along the phylogeny compared to DR and ClaDS. Conversely, DR and ClaDS recover higher rate heterogeneity compared

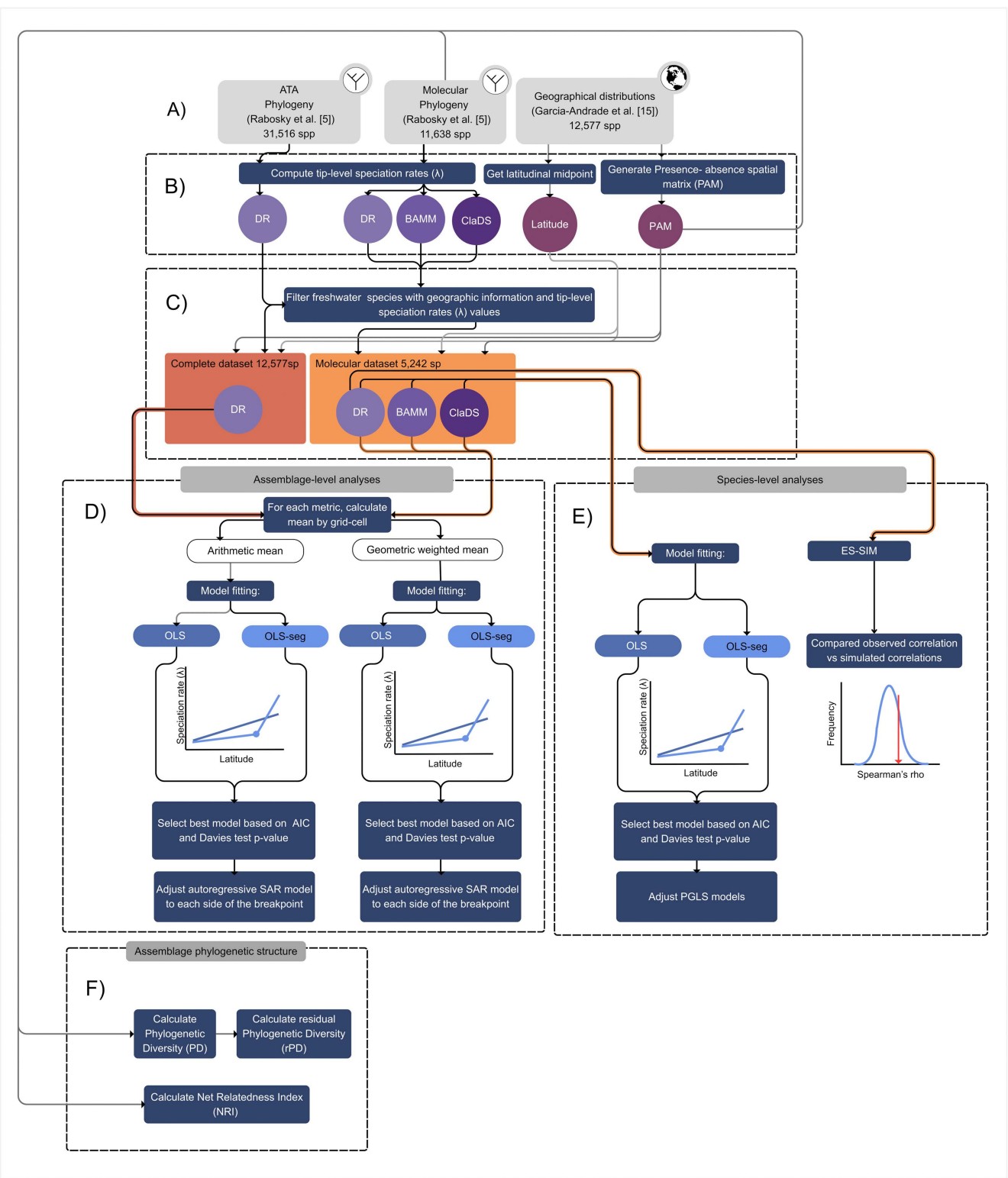

**Fig 1. Flowchart depicting the methodological workflow for this study. (A)** Geographic and phylogenetic data sources. **(B)** Compute speciation rates and geographic information (midpoint latitude and presence-absence matrix (PAM)). **(C)** Create final data sets by filtering freshwater species with geographic information and tip-level speciation rates (λ) values. **(D)** Assemblage-level analyses to evaluate the relationship between latitude and speciation rates. **(E)** Species-level analyses to evaluate the relationship between latitude and speciation rates. **(F)** Analysis of phylogenetic assemblage structure.

to BAMM, with ClaDS being able to identify many small and frequent shifts that result in more heterogeneous rates among lineages [12]. Accordingly, considering these three metrics allowed us to evaluate the robustness of our studied patterns beyond methodological differences.

After calculating speciation rates for all Actinopterygians fishes, we focused on freshwater species, defined as all those inhabiting freshwater and/or brackish ecosystems. Considering both phylogenies (completed ATA phylogenies and the molecular phylogeny) along with the available geographic ranges, we obtained two final datasets: i) a "complete" dataset with 12,557 species, representing 77% of freshwater fishes with geographic ranges and λDR speciation rate data; and ii) a "molecular" dataset with 5,242 freshwater species from the molecular phylogeny, representing ~30% of all freshwater fish species described to date with geographic ranges and λDR, λTC, λTV and λClaDS speciation rate values (Fig 1C). Then, we correlated the values between each pair of these metrics to evaluate their consistency.

### Assemblage-level analyses

To reconstruct the latitudinal speciation gradient for freshwater fishes at the assemblage level for both datasets, complete and molecular, we first created a species' presence-absence matrix using a global grid-cell system of 111 x 111 km resolution overlaid onto the projected geographic ranges using the letsR package [43]. This resolution is recommended when using geographic ranges to represent biodiversity gradients at large spatial scales, as they provide similar patterns than those based on occurrences but allowing larger coverage of species and spatial extent [44]. Importantly, recent studies on fish macroecology such as those describing the LDG of freshwater fishes [15] and the latitudinal speciation gradient of marine fishes [5] have used this same resolution and based their analyses on geographic ranges, allowing us to discuss our findings in light of these relevant studies. For each grid-cell assemblage of co-occurring species, we computed the mean speciation rate value for each considered metric in two different ways: (i) arithmetic mean of all species in each grid cell, and (ii) the geometric mean of all species in each grid-cell weighted by the inverse of their range size (Table S1.1 in S1 Appendix). Species weights were determined as the inverses of the number of grid cells in which each species was present (Table S1.1 in S1 Appendix). These metric means were employed to mitigate biases arising from the sensitivity of the arithmetic mean to extreme values and to control the influence of widely distributed species [5].

We assessed the relationship between assemblage speciation rates and latitude applying two models, with the speciation rate as response variable and the absolute latitude as predictor variable. First, we employed ordinary least squares (OLS) to model a linear relationship followed by a segmented OLS models (OLS-seg) to evaluate the presence of a breakpoint in the studied relationship, corresponding to different slopes for distinct regions of the latitudinal gradient. We selected the best model based on AIC, followed by a Davies-test to determine the significance of the breakpoint. For this, we used the segmented package for R [45]. If the best model was OLS-segmented, separate OLS and Spatial Autoregressive (SAR) models were fitted for each region before and after the breakpoint and compared based on AIC. These SAR models were fitted to consider the presence of spatial autocorrelation. We evaluated two neighborhood distance matrices among the grid-cell centroids, 111 km (minimum) and 1000 km (maximum), and three weighting schemes for the distance matrix: row-standardized (W), globally standardized (C), and a variance-stabilizing coding scheme (S) [46]. We chose the best-fitting SAR model based on AIC values and the highest $R^2$. SARs models were fitted using the spatialreg package [47] in R (Fig 1D).

### Species-level analyses

To evaluate the latitude-speciation relationship at the species level, we only considered the species present in the molecular phylogeny. We calculated the absolute latitudinal midpoint for each species and related them to their speciation rates using OLS, and OLS-segmented models for each speciation metric. Additionally, we conducted phylogenetic regressions (PGLS) to consider the shared evolutionary history among species and avoid high type 1 error rates on regression

coefficients [48]. We compared the fit of these models based on AIC. Finally, we ran an ES-SIM model to test the effect of species traits (in this case, absolute latitudinal midpoint) on their speciation rates considering only the λDR metric. This method is robust to model misspecification and phylogenetic pseudo-replication [49] (Fig 1E). All models that relate (absolute) latitude and speciation rate were fitted using latitude in meters to conform with the use of projected range maps. However, to facilitate depicting observed patterns and discussing our findings, we report our results with latitude in degrees.

### Assemblage phylogenetic structure

To aid the interpretation of the latitudinal speciation gradient and infer potential underlying processes, we evaluated the phylogenetic structure of freshwater fish assemblages across the globe. We calculated three metrics per grid-cell assemblage using the ATA phylogenies: phylogenetic diversity (PD), residual phylogenetic diversity (rPD), and Net Relatedness Index (NRI) (Fig 1F). PD considers the shared ancestry among species and is quantified as the total branch length of the tree that connects all species present within an assemblage, which is inherently positively correlated to species richness [50]. rPD is the phylogenetic diversity that is not explained solely by species richness and can be used as a proxy of how evolutionary events (i.e., speciation, extinction, and dispersal) have contributed to species assemblages, where areas with low rPD harbor less evolutionary history than expected from their species richness resulting from high *in situ* speciation and few dispersal events, whereas areas with high rPD harbor more evolutionary history than expected from species richness resulting from low *in situ* speciation and frequent dispersal events [31]. We estimated rPD by extracting the residuals of a locally estimated scatterplot smoothing (LOESS) where species richness was predicted by PD. Finally, NRI measures the standardized effect size of the mean pairwise distance of all species conforming an assemblage relative to random assemblages from a species pool [51]. Values of NRI above 1.96 indicate phylogenetic clustering (i.e., more closely related species than expected by chance). Negative values below −1.96 indicate phylogenetic overdispersion (i.e., more distantly related species than expected by chance), and values between 1.96 and −1.96 indicate no (i.e., random) phylogenetic structure relative to the species pool [51]. We defined the species pool for each grid-cell assemblage as the set of species present in their corresponding biogeographic regions following [28]: Nearctic, Neotropical, Palearctic, Afrotropical, Australian, Indo-Malay, and Oceania.

## Results

Both datasets, complete (12,557 species) and molecular (5,242 species), exhibited the same geographic pattern following the classic LDG, with higher species richness in the tropics compared to the temperate regions (Fig 2 for the molecular set and Fig S1.1 in S1 Appendix for the complete set). Regarding speciation rates, the average tip-level speciation rate in the complete dataset was λDR = 0.21 species per Myr. In the case of the molecular dataset, mean speciation rates showed similar values for λDR and λClaDS metrics (0.21, 0.17; respectively), whereas values for the BAMM-derived metrics were (λTC = 0.23, λTV = 0. 21). Despite methodological differences among the speciation rate metrics, we found a high correlation (Pearson's r > 0.8) between all the metrics at the species level (Fig S1.2 in S1 Appendix). Accordingly, in the next sections, we only present results for λDR, λClaDS, and λTC, thus excluding one of the BAMM-derived metrics, λTV, whose results can be found in the extended appendix.

### Latitudinal speciation gradient at the assemblage level

At the assemblage level, mean speciation rate across grid-cells in the complete dataset was λDRw = 0.14, while in molecular dataset these values were higher for the BAMM-derived metric (λTCw = 0.19) and lower for the λDRw (0.13) and λClaDSw (0.10). All speciation metrics showed strong geographic structure, with similar peaks of speciation rate values around the tropics (11°-13° south, 11°-12° north) and at northern latitudes (>45°) (Fig 2). Despite the tropics exhibiting

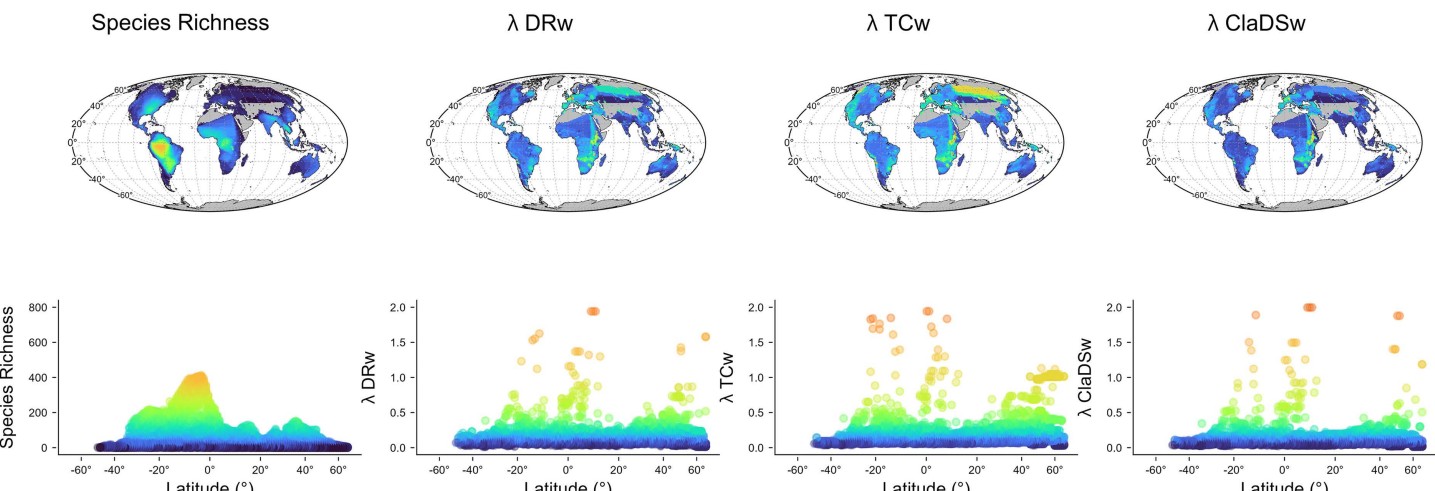

**Fig 2. Geographic pattern of species richness and speciation rates.** Patterns under different speciation rate metrics for the molecular dataset of 5,242 freshwater fish species. The upper row depicts the geographic patterns and the lower row their bivariate representations. Maps projection is in Mollweide equal area. The maps were created by the authors in R using the open-source "rnaturalearth" package [52] with public domain data from Natural Earth (http://www.naturalearthdata.com/). The figure is published under the CC BY 4.0 license.

the highest speciation values, our results consistently revealed a positive latitudinal gradient in which overall speciation rates increased towards higher latitudes outside the tropics, mainly in the northern hemisphere (Fig 1). Speciation rates also display an inverse relationship with species richness, such that regions with lower richness showed higher speciation rates (Fig 3, Fig S1.3 in S1 Appendix).

The global OLS model showed a significant positive relationship between absolute latitude and speciation rate using the complete dataset and the λDRw metric (β = 0.0003, $R^2$ = 0.0029, p < 0.001). When considering the molecular dataset, the OLS models showed a significant positive relationship between latitude and speciation rate for λTCw (β = 0.0033 $R^2$ = 0.0652, p < 0.001) and negative for λClaDSw (β = −0.0002, $R^2$ = 0.0012, p < 0.05), whereas the relationship was not significant for λDRw (Fig 4, Table S2.1 in S2 Appendix). The other variants of the speciation rate metrics showed positive and significant patterns, except λClaDS that showed a non-significant relationship (Table S2.1 in S2 Appendix).

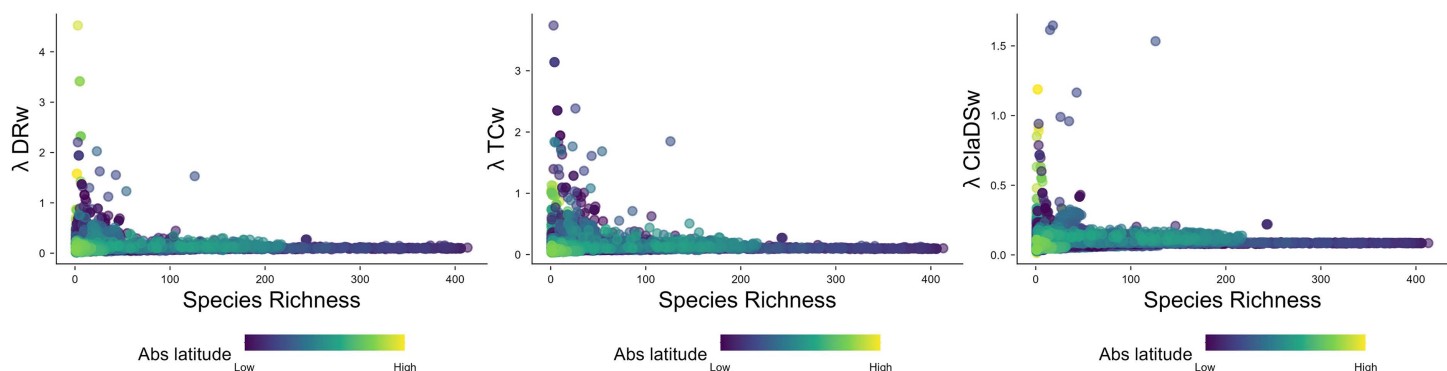

**Fig 3. Relationship between species richness and speciation rates.** Bivariate plots for the molecular dataset of 5,242 freshwater fish species depicting the relationship between species richness and speciation rates for the three metrics: λDRw, λTCw, λClaDSw.

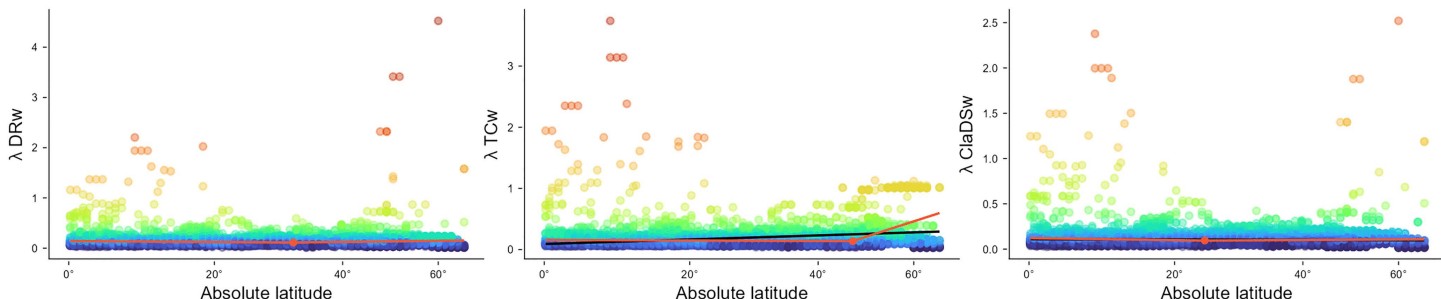

**Fig 4. Assemblage-level relationship between speciation rate and the absolute value of latitude.** Regressions considering different adjusted models (black – OLS model, red – OLS segmented). All X-axes are in degrees only for illustrative purposes. See Fig S2.1 to S2.2 in S2 Appendix for more detailed depictions of the statistical relationships.

OLS-segmented models provided a better fit than simple OLS models and identified a single breakpoint in the speciation-latitude relationship for all metrics. The location of these breakpoints along the latitudinal gradient differed among metrics and datasets, varying between 24.39° and 47.83°. In the complete data set, the latitudinal breakpoint for λDRw was 47°, while in the molecular data set the values were λTCw = 46.37°, followed by the λDRw = 31.8° and λClaDSw = 24.39° (Table S2.1, Fig S2.1-S2.2 in S2 Appendix).

When fitting models separately for each region around the identified breakpoints, in general, the left-side models (i.e., below the breakpoint) showed negative relationships in OLS models and non-significant relationships between speciation and latitude in SAR models (Table S2.2 in S2 Appendix). OLS models showed negative relationships for λDRw complete ($\beta = -0.0004$, $R^2 = 0.0043$, $p < 0.001$) and in molecular λDRw ($\beta = -0.0005$, $R^2 = 0.003$, $p < 0.001$), λTCw ($\beta = -0.0004$, $R^2 = 0.0012$, $p < 0.05$) and λClaDSw ($\beta = -0.001$, $R^2 = 0.004$, $p < 0.001$). In contrast, both OLS and SAR models fitted to the right-side of the breakpoint (i.e., above it) consistently showed significant and positive relationships between speciation and latitude across all speciation metrics (SAR results where λDRw complete: $\beta = 0.0079$, $R^2 = 0.2926$, $p < 0.001$; λDRw molecular: $\beta = 0.0043$, $R^2 = 0.2189$, $p < 0.05$; λTCw: $\beta = 0.0234$, $R^2 = 0.83$, $p < 0.001$; λClaDSw: $\beta = 0.0006$, $R^2 = 0.6138$, $p < 0.001$, Tables S2.3 in S2 Appendix). Overall, the relationship between speciation and latitude across freshwater fish assemblages holds true only at higher latitudes outside the tropics. All SAR models fitted to the right-side of the breakpoint showed higher effect sizes and steeper slopes for the BAMM-derived metric λTC, followed by the λDR and the λClaDS metrics (Table S2.3 in S2 Appendix).

## Latitudinal speciation gradient at the species-level

At the species level, considering only the molecular dataset, the latitudinal speciation gradient was somewhat similar to the assemblage-level pattern, with species located at the tropics and at northern latitudes exhibiting higher speciation rates than species with midpoints at other latitudes. The highest speciation rates at latitudes >45°, namely in the northern hemisphere, belonged to species from the Salmoniformes, Perciformes, Clupeiformes, and Cypriniformes which also have marine counterparts (Fig S3.1 in S3 Appendix), whereas in the tropics these belonged to species of the order Cichliformes. Notably, prominent groups of Ostariophysi (Characiformes and Siluriformes) exhibited similar speciation rates along the latitudinal gradient (Fig 5, Fig S3.2 in S3 Appendix).

In contrast to the assemblage-level models, OLS models at the species-level showed significant negative relationships between speciation and latitude for all metrics, but with low coefficients for both the slope and effect sizes (λDR: $\beta = -0.0008$, $R^2 = 0.0009$, $p < 0.05$; λTC: $\beta = -0.0011$, $R^2 = 0.002$, $p < 0.05$; λClaDS: $\beta = -0.0011$, $R^2 = 0.0026$, $p < 0.001$) and no significant breakpoints were found in the OLS-segmented models (Table S3.1 in S3 Appendix). Finally, PGLS models showed the best fit across models, with a significant positive relationship only for λDR with high phylogenetic signal

**Fig 5. Species-level relationship between speciation rates and latitude.** Phylogenetic (top panel) and latitudinal (bottom panel) patterns of tip-level speciation rates. Patterns are shown for λDR, λTC, and λClaDS. Next to the phylogeny, we show the speciation rate per species at their corresponding midpoint along the latitudinal gradient (x-axis). Species have the same colors in both panels. The silhouettes of fishes were taken from phylopic ("https://www.phylopic.org/").

(β = 0.0015, R² = 0.00124, p < 0.05; Pagel's λ > 0.75, Table S3.1, Fig S3.3 in S3 Appendix). However, the ES-SIM analysis did not support a significant relationship between species' latitudinal position and their speciation rates (rho = 0.0225, p = 0.89).

## Assemblage phylogenetic structure

As expected, the geographic pattern of phylogenetic diversity (PD) closely followed that of species richness. However, when accounting for the effect of species richness via the residual PD (rPD), a distinct geographic pattern emerged. The lowest rPD values were located in the Afrotropic central region, particularly in Lakes Malawi, Tanganika and Victoria. Negative rPD were also present in some regions from southern North America to South America, western Europe, east Asia, and northern Oceania. Conversely, positive rPD values were present in northern Africa, central India, South Asia, the Malay Archipelago, and northern Siberia (Fig 6).

Within biogeographic regions, most assemblages showed no phylogenetic structure (non-significant NRI values), with the notable exception of the Neotropical and Nearctic regions (Fig 6). In the Neotropical region, most assemblages across central and northern South America showed positive NRI values, indicating the co-occurrence of closely related species. Still, the highest positive NRI values were found in the Afrotropical region, namely in the assemblages corresponding to Lakes Malawi, Tanganyika, and Victoria. In this region, mainly in the equatorial part, there were several assemblages with negative NRI values, indicating the presence of distantly related species. In the Nearctic region, there was a longitudinal pattern with assemblages showing negative NRI values around the central part of this region, whereas assemblages with positive NRI values were located towards the western and southern parts of this region.

## Discussion

Here, we evaluated the geographic variation of speciation rates in freshwater fishes on a global scale. To our knowledge, this is the first study to focus exclusively on how speciation rates vary across latitude at both assemblage and species levels using species-specific geographic ranges instead of basin-level species lists. Overall, our assemblage-level results revealed a positive relationship between latitude and speciation rates, with speciation rates increasing towards higher latitudes, namely in the northern hemisphere, despite the tropics exhibiting the highest single speciation values. Species-level results did not show a significant relationship between speciation rates and latitude. However, both assemblage- and species-level approaches combined with the assemblage phylogenetic structure analysis allowed us to conduct a more comprehensive evaluation of the latitudinal speciation gradient in freshwater fishes revealing region-specific effects across northern latitudes as well as clade-specific patterns.

Our observed assemblage-level pattern aligns with recent studies for various taxa, where speciation rates are positively related with latitude, including birds [17], mammals [13], plants [53], and marine fishes [49]. Along with these studies, our findings add evidence that the classic LDG does not necessarily result from higher speciation rates in the tropics. In fact, recent studies on freshwater fishes have found that evolutionary time (i.e., the time elapsed since colonization/origination that allows species accumulation) is the primary driver of species richness at the class [4] and order level [15]. In addition, other evolutionary factors such as low extinction rates in the tropics have been found as mainly responsible for the LDG in other vertebrate groups (e.g., mammals [54]). Although freshwater fish speciation rates followed an inverse latitudinal gradient relative to species richness somewhat similar to the pattern observed for marine fishes [49], there are important differences in the species and assemblage level patterns between these two groups of fishes.

The latitudinal speciation gradient of freshwater fishes was less consistent than that observed for marine fishes. In marine fishes, Rabosky et al. [5] found a consistently positive global relationship between latitude and speciation, regardless of their considered metrics (BAMM or DR). In contrast, the global relationship for freshwater fishes was not consistent between these two metrics, with BAMM showing the same pattern to marine fishes whereas DR showed no significant latitudinal trend. Indeed, marine fishes exhibited a clear u-shaped pattern between latitude and speciation, with high speciation rates at both northern and southern latitudes [5]. Instead, freshwater fishes did not show such a clear pattern but a more heterogenous one with higher variation of speciation rates in tropical regions (Fig 2). This difference between marine and freshwater fishes is mainly given by tropical freshwater assemblages exhibiting very high speciation rates, whereas tropical marine assemblages showed consistently low speciation rates [5]. Our observed speciation peaks in the tropics

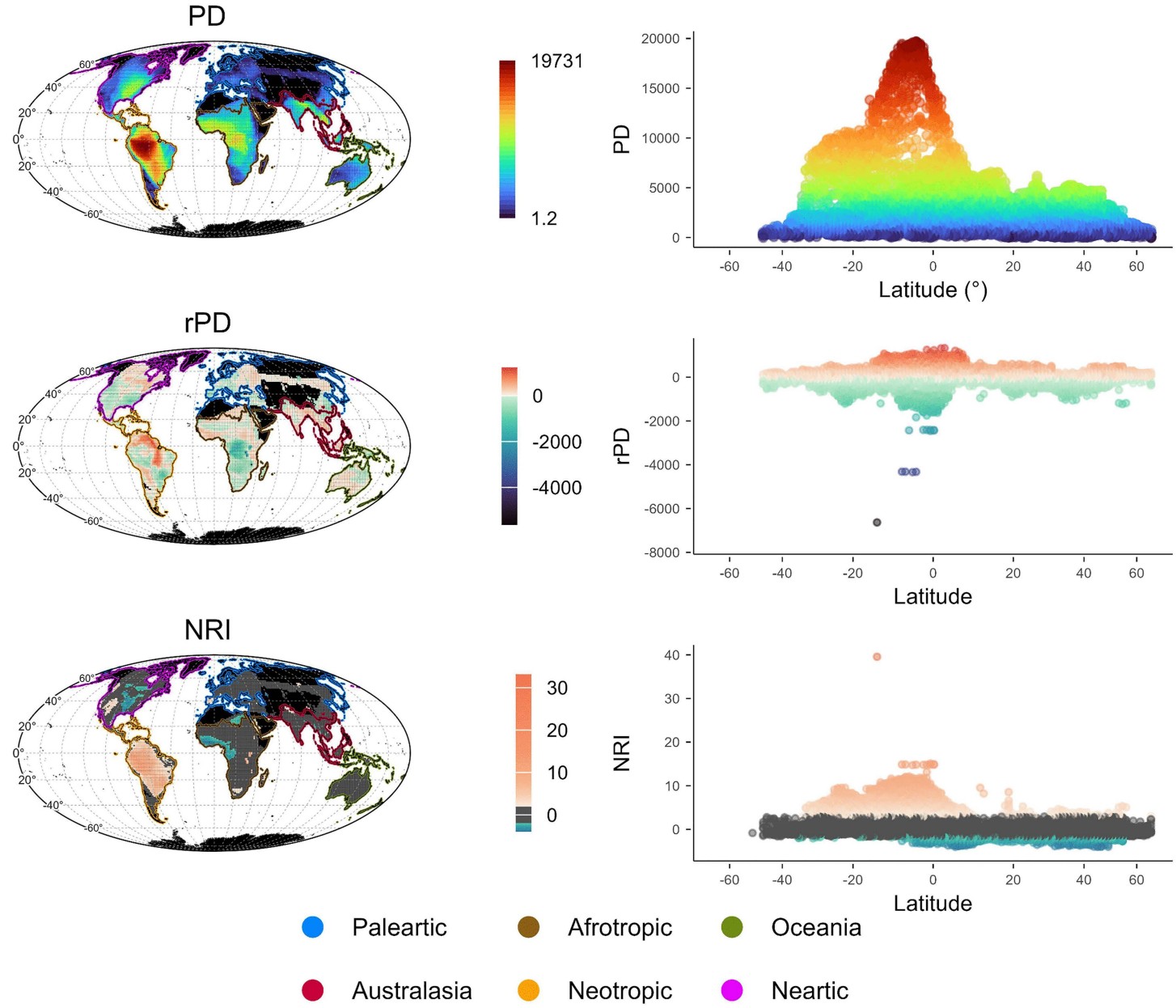

**Fig 6. Geographic patterns of phylogenetic structure.** From top to bottom, phylogenetic diversity (PD), residual phylogenetic diversity (rPD), and (NRI) Net Relatedness Index of freshwater fish assemblages across the globe. Maps projection is in Mollweide equal area. The maps were created by the authors in R using the open-source "rnaturalearth" package [52] with public domain data from Natural Earth (http://www.naturalearthdata.com/). The figure is published under the CC BY 4.0 license.

were located in places with high phylogenetic clustering, indicating the co-occurrence of recently derived and/or closely related lineages within the assemblages (Fig 6), which is consistent with recent findings for freshwater fishes at the basin level [4], where the highest speciation rates are located in recently colonized basins. Assemblages with high phylogenetic clustering and high speciation rates in tropics are mainly found in Africa, especially in the lakes Malawi, Tanganyika, and Victoria, where adaptive radiation driven by ecological opportunities for specialization has resulted in high recent

speciation in the Cichliformes group approximately 15 Myr ago [55]. Another example is the Congo Basin, where hyper-diverse assemblages of Mormyridae (Osteoglossiformes) diversified around 40 Myr [56]. Similar processes have been suggested in South America, namely in La Plata Basin, where the uplift of the Serra do Mar and Serra da Mantiqueira created isolated habitats and opportunities for vicariance, particularly among upland species like *Hypostomus* (Siluriformes) 30 Myr ago [33,57]. In the same vein, fragmentation has driven speciation in the western upland Andean regions in groups such as *Orestias* [57]. Conversely, marine fish do not exhibit such peaks in speciation; indeed, the highest rates of speciation in marine environments only occur outside the tropics [5]. This leads to the identification of a narrower interval for the latitudinal breakpoint in segmented models for marine fishes, between 27° and 41° latitude [5], compared to freshwater fishes where the latitudinal position of the breakpoint was broader, ranging between 24.39° and 47.83°, depending on the speciation rate metric. In general, freshwater fish speciation showed no significant latitudinal pattern below our identified breakpoint (i.e., towards lower latitudes), whereas above the breakpoint (i.e., towards higher latitudes) the relationship between latitude and speciation becomes positive and significant across all Actinopterygians, as shown here for freshwater fishes and by [5] for marine fishes.

In marine fishes, higher speciation outside the tropics has been explained by increased opportunities to transition between different depths [58] and rapid expansion into new habitats over the last 80 Kya, following repeated glaciation and deglaciation events that wiped out local fauna [59,60]. These events likely fostered speciation through adaptive radiation and ecological opportunities. Indeed, in marine fishes, high-speciation areas have been found in both southern and northern hemispheres, which coupled with morphological evolution provides evidence of radiation in both north and south regions outside the tropics [58,59]. Conversely, our findings for freshwater fish revealed that the relationship between latitude and speciation beyond the breakpoint is driven by assemblages located in the northern hemisphere, with steeper slopes compared to those obtained from the segmented models of marine fishes [5]. This outcome could result from the greater availability of habitat at high latitudes in the northern hemisphere, which comprises 74% of the Earth's landmass [61]. As a result of this geometric constraint, the southern hemisphere has fewer available areas for freshwater fishes, limiting opportunities for dispersal and colonization in southern regions.

Freshwater fish assemblages with high speciation rates in the Nearctic and Palearctic regions, particularly below 45° latitude, showed negative rPD and positive NRI values, indicating recent radiation and phylogenetic clustering. This suggests region-specific effects with these areas serving as refugia during glacial periods, where climatic stability and time allowed for speciation and the accumulation of species, as previously proposed for North American fishes [62]. The high speciation values in these areas could be explained by allopatric speciation mediated by dispersal [63] or by adaptive radiation, particularly in lakes that offer greater ecological opportunities [55]. This is evident in genera such as *Coregonus* and *Salvelinus* (Salmoniformes), which underwent rapid adaptive radiation prior to the Last Glacial Maximum (80–90 kya) [64,65] as well as in *Cottus* (Perciformes) within Lake Baikal [66]. Adaptative radiation could also have occurred in streams, as observed in North America Cypriniforms that diversify in the benthic-to-pelagic axis [67]. At even northern latitudes, above 45°, assemblages showed higher speciation rates and positive rPD values, indicating frequent dispersal events and/or environmental filtering [31]. This pattern is likely due to glaciations and sea-level fluctuations in these areas, which have repeatedly reshaped habitats and driven species to colonize new environments [62] following disturbance events [68]. Indeed, species in these assemblages often exhibit high tolerance and dispersal capabilities. For example, species from the order Salmoniformes, which originated in the northern Pacific Ocean, evolved to become anadromous and subsequently migrated to the polar basin and North Atlantic [69], as well as Gobiiformes that probably originated in freshwater environments of this region and show multiple transition to marine and euryhaline habitats [70]. As such, the northernmost assemblages (above 45°) with higher speciation rates could have resulted from the dispersal of highly radiating lineages from lower latitudes (below 45°).

Despite revealing a consistent pattern of higher speciation rates towards higher latitudes at the assemblage level, the location of the latitudinal breakpoint varied depending on the speciation rate metric. This can be related to the metrics'

methodological differences. For example, BAMM-derived (e.g., λTC) estimates identify more discrete shifts along the phylogeny compared to both DR and ClaDS [11,12], which can explain its higher latitudinal breakpoint as mainly driven by particular clades with large diversification shifts that belong to the same macroevolutionary regime (e.g., Salmoniformes; [39]). In contrast, DR and ClaDS estimates provide more heterogeneous rates as they accommodate higher rate variation along the phylogeny, especially ClaDS that is capable of identifying many small and frequent shifts in diversification regimes [42]. As a result, under DR and mainly ClaDS, more lineages beyond those belonging to large and discrete macroevolutionary regimes show high speciation rates and push the breakpoint to lower latitudes compared to BAMM given their particular geographic distributions extending to such latitudes (e.g., Cypriniformes).

At the species level, we did not recover a significant latitudinal gradient, although some species did exhibit high (but not the highest) speciation rates in northern latitudes. This discrepancy compared to the assemblage-level pattern may arise from our use of geographic range centroids as proxies for species latitudinal positions, which inherently reduce the spatial variation of observed patterns. More importantly, this finding results from clade-specific effects such as the disproportionate influence of the hyper-diverse and regionally concentrated Cichliformes in tropical zones, which comprise several species with the highest speciation rates (Fig 5). Indeed, despite other groups showing high speciation rates outside the tropics, such as Salmoniformes that are restricted to the northern hemisphere and the more widely distributed Cypriniformes and Clupeiformes, the overall species-level pattern is balanced by clades with high speciation rates in both tropical and temperate latitudes ultimately leading to the absence of a latitudinal gradient in speciation rates. This finding was consistent among speciation rate metrics, which at the species-level showed high correlations (Pearson's r > 0.8) perhaps due to the strong signal left by the high-speciating clades that is consistently picked up by all metrics despite their methodological differences and thus leading to their concordance. In contrast, this balancing between tropical and temperate high-speciating clades at the species-level does not erase the latitudinal gradient observed at the assemblage-level. This is mainly because the effect of high-speciating Cichliformes lineages in tropical latitudes is countered by the high species richness and thus co-occurrence of many other lineages from other clades exhibiting lower speciation rates at these same latitudes, whereas the high-speciating lineages at the northern temperate regions co-occur with a lower number of lineages that on average also have high speciation rates (Fig 3).

Taken together assemblage and species level analysis, the findings suggest there are a regional-effect beyond the clade-specific patterns revealing historical contingencies and environmental conditions across the Northern Hemisphere that promoted high species rates. Still, this invites caution interpretation without ignoring clade-specific (non-replicated) factors that generate this pattern [25]. For example, LDG in freshwater fishes exhibits a classic unimodal richness pattern, but this pattern at the class level emerges from patterns exhibited by individual orders that often display different species richness gradients such as unimodal, bimodal, or multiple peaks [15]. Indeed, a pattern deconstruction approach to evaluate the latitudinal speciation gradient of freshwater fishes at different taxonomic/phylogenetic levels (e.g., orders, [15]) seems necessary to reveal general or idiosyncratic drivers behind such gradient.

Although we found consistent patterns robust to methodological differences of speciation rate metrics, there are other aspects that could bias our results. In particular, the pervasive knowledge shortfalls that hinder our understanding of biodiversity patterns in general [71] and those of fishes in particular [72,73]. For example, it is well known that taxonomic diversity is better documented in temperate regions than in tropical regions [74], thus a higher Linnean shortfall in the tropics that is particularly acute for Neotropical fishes given the high number of potential undescribed species [21,75]. Combined with the Darwinian shortfall, regarding our incomplete information on the evolutionary relationships among species [71] and recently recognized for fishes [76] speciation rates in the tropics may in fact be higher than the ones documented in our study [77]. In the same vein, the Wallacean shortfall regarding incomplete knowledge on species' geographic distributions [71] can further obscure patterns mainly at the assemblage level where estimates are derived from the complete ranges of all species across space. Notwithstanding these shortfalls, we found consistent patterns between our two data sets, one considering about 78% (12,557 species) of all described species of freshwater Actinopterygii (the complete

dataset) and the molecular dataset with less than half (5,242) of all freshwater fish species. Continued taxonomic revision, expanded geographic sampling, and more refined phylogenetic hypotheses will certainly warrant a reevaluation of our findings. Therefore, our study provides a valuable baseline for future comparative studies as biodiversity knowledge continues to improve.

## Conclusions

By integrating assemblage- and species-level approaches, our study revealed an inverse latitudinal gradient of speciation relative to species richness in freshwater fishes only at assemblage level, with speciation rates increasing towards higher latitudes, namely in the northern hemisphere, even when the highest rates were located in the tropics. This pattern aligns with similar trends observed in terrestrial vertebrates and marine fishes. Overall, the areas with high speciation rates are congruent with areas of phylogenetic clustering and recent radiation, whereas the northern temperate areas with high speciation show evidence of recent radiation and high clustering as well as dispersal and environmental filtering that further align with glaciation dynamics. These findings suggest the effect of past climatic events and refuge areas as well as biogeographic and evolutionary factors in shaping speciation gradients. Further research should evaluate these (and potentially other) factors to fully understand the drivers behind the geographic variation of speciation rates.

## Supporting information

**S1 Appendix. Additional description of methods, metrics and datasets used in this study.**
(DOCX)

**S2 Appendix. Results from the assemblage-level analysis.** Tables and figures corresponding to the assemblage-level results.
(DOCX)

**S3 Appendix. Results from the species-level analysis.** Tables and figures corresponding to the species-level results.
(DOCX)

## Acknowledgments

JH-P thanks the doctoral program of the Instituto de Ecología, A.C. (INECOL) and the support of the Consejo Nacional de Humanidades, Ciencias y Tecnologías (CONAHCYT, México. The authors gratefully acknowledge the computer resources, technical expertise, and support provided by the Laboratorio Nacional de Supercómputo del Sureste de México, a CONACYT member of the network of national laboratories, and the Huitzilin Cluster team at INECOL for their invaluable assistance, especially Emanuel Villafán and Abraham Vidal-Limón for providing us with the computer resources and support to run our analysis. We also sincerely appreciate all the researchers who contributed to developing new methods and releasing collected data that allowed us to conduct this study. We thank SSB and UNAM for organizing the SSB 2023 Standalone Meeting as well as the German Centre for Integrative Biodiversity Research (iDiv) Halle-Jena-Leipzig, for providing platforms to discuss and refine the ideas behind this study.

## Author contributions

**Conceptualization:** Juliana Herrera-Pérez, Juan Carvajal-Quintero, Fabricio Villalobos.

**Data curation:** Juliana Herrera-Pérez, Juan Carvajal-Quintero, Axel Arango, Daniel Valencia-Rodríguez, Ana Berenice García-Andrade, Pablo Tedesco.

**Formal analysis:** Juliana Herrera-Pérez, Axel Arango, Fabricio Villalobos.

**Investigation:** Juliana Herrera-Pérez, Juan Carvajal-Quintero, Pablo Tedesco, Fabricio Villalobos.

**Methodology:** Juliana Herrera-Pérez, Axel Arango, Daniel Valencia-Rodríguez, Ana Berenice García-Andrade, Pablo Tedesco, Fabricio Villalobos.

**Project administration:** Juan Carvajal-Quintero, Fabricio Villalobos.

**Resources:** Juan Carvajal-Quintero, Axel Arango, Daniel Valencia-Rodríguez, Ana Berenice García-Andrade, Pablo Tedesco, Fabricio Villalobos.

**Software:** Juliana Herrera-Pérez, Ana Berenice García-Andrade.

**Supervision:** Juan Carvajal-Quintero, Fabricio Villalobos.

**Validation:** Juliana Herrera-Pérez, Juan Carvajal-Quintero, Pablo Tedesco, Fabricio Villalobos.

**Visualization:** Juliana Herrera-Pérez, Juan Carvajal-Quintero.

**Writing – original draft:** Juliana Herrera-Pérez, Fabricio Villalobos.

**Writing – review & editing:** Juan Carvajal-Quintero, Axel Arango, Daniel Valencia-Rodríguez, Ana Berenice García-Andrade, Pablo Tedesco, Fabricio Villalobos.

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
