## [Decision Letter · Decision Letter 0]

29 Aug 2025

Dear Dr. Villalobos,

Thank you for submitting your manuscript to PLOS ONE. After careful consideration, we feel that it has merit but does not fully meet PLOS ONE’s publication criteria as it currently stands. Therefore, we invite you to submit a revised version of the manuscript that addresses the points raised during the review process.

We look forward to receiving your revised manuscript.

Kind regards,

Tzen-Yuh Chiang

Academic Editor

PLOS ONE

Journal Requirements:

Consejo Nacional de Humanidades, Ciencias y Tecnologías (CONAHCYT, México) (doctoral scholarship #1149672). German Centre for Integrative Biodiversity Research (iDiv) Halle-Jena-Leipzig (German Research Foundation FZT-118 grant 202548816) via sDiv and FLEXPOOL.

JH-P thanks the doctoral program of the Instituto de Ecología, A.C. (INECOL) and the support of the Consejo Nacional de Humanidades, Ciencias y Tecnologías (CONAHCYT, México) for providing the doctoral scholarship (1149672). The authors gratefully acknowledge the computer resources, technical expertise, and support provided by the Laboratorio Nacional de Supercómputo del Sureste de México, a CONACYT member of the network of national laboratories, and the Huitzilin Cluster team at INECOL for their invaluable assistance, especially Emanuel Villafán and Abraham Vidal-Limón for providing us with the computer resources and support to run our analysis. We also sincerely appreciate all the researchers who contributed to developing new methods and releasing collected data that allowed us to conduct this study. Furthermore, we thank SSB and UNAM for organizing the SSB Standalone Meeting, which provided a platform to discuss and refine the ideas behind this study. JDC-Q and JH-P were supported by the German Centre for Integrative Biodiversity Research (iDiv) Halle-Jena-Leipzig (German Research Foundation FZT-118 grant 202548816) via sDiv and FLEXPOOL.

Consejo Nacional de Humanidades, Ciencias y Tecnologías (CONAHCYT, México) (doctoral scholarship #1149672). German Centre for Integrative Biodiversity Research (iDiv) Halle-Jena-Leipzig (German Research Foundation FZT-118 grant 202548816) via sDiv and FLEXPOOL.

6. We note that Figures 1, 4, S1- 1, S2 -1, S2 -2, S2 -3, S2 -4, and S2 -5 in your submission contain map images which may be copyrighted. All PLOS content is published under the Creative Commons Attribution License (CC BY 4.0), which means that the manuscript, images, and Supporting Information files will be freely available online, and any third party is permitted to access, download, copy, distribute, and use these materials in any way, even commercially, with proper attribution. For these reasons, we cannot publish previously copyrighted maps or satellite images created using proprietary data, such as Google software (Google Maps, Street View, and Earth). For more information, see our copyright guidelines: http://journals.plos.org/plosone/s/licenses-and-copyright.

a. You may seek permission from the original copyright holder of Figures 1, 4, S1- 1, S2 -1, S2 -2, S2 -3, S2 -4, and S2 -5 to publish the content specifically under the CC BY 4.0 license.

Reviewers' comments:

Reviewer's Responses to Questions

**Comments to the Author**

1. Is the manuscript technically sound, and do the data support the conclusions?

Reviewer #1: Partly

Reviewer #2: Partly

Reviewer #3: Partly

2. Has the statistical analysis been performed appropriately and rigorously?

Reviewer #1: No

Reviewer #2: Yes

Reviewer #3: Yes

3. Have the authors made all data underlying the findings in their manuscript fully available?

Reviewer #1: Yes

Reviewer #2: Yes

Reviewer #3: Yes

4. Is the manuscript presented in an intelligible fashion and written in standard English?

Reviewer #1: Yes

Reviewer #2: Yes

Reviewer #3: Yes

Reviewer #1: Review of PONE-D-25-26611

The latitudinal speciation gradient in freshwater fishes: higher speciation at higher latitudes in the northern hemisphere

Herrera-Pérez et al.

This paper examines patterns of speciation rates of freshwater fishes across latitude, globally, finding that speciation rates increased with latitude – mainly in the northern hemisphere. The aim of the study was to describe and evaluate the latitudinal pattern of speciation in freshwater fish. And, also to determine whether speciation rates were related/responsible for the global pattern of species richness – higher richness at the equator � but this was not explicitly stated. Speciation was evaluated at the assemblage- and species-level using various metrics of tip-level speciation rate and multiple models. Findings indicated the importance of glaciations/refugia, biogeography, and evolutionary factors on speciation gradients. Where speciation was high, rates were congruent with recent radiation events of specific taxonomic groups; dispersal and environmental filters related to glaciation events likely played an important role here. This is an interesting paper, which highlights the importance of fragmentation and environmental filters on diversity.

The material in the introduction and discussion is well written and clear with supporting citations. My main criticism – albeit a large one – is that the methods are unruly, hard to follow, and poorly justified. I think much of this can be corrected, and as clarifications are made, I believe some of the other more minor questions I had in the discussion will be addressed.

I think the paper would benefit substantially from a flow chart that demonstrates how all of data and the models came together and how choices were/are made. There are numerous streams of information including two pattern levels, two datasets, multiple metrics of the tip-level speciation rate, several types of model structure, multiple mean calculations, and transformations. This is very complicated, and I cannot make sense of why certain things were done (e.g., the use of GAMs in addition to the other model types).

Instead of running all the models on all the metrics and showing all the results, I suggest narrowing the focus. What is the main question you are asking in this paper? What model (GAM, breakpoint, …) best fits the data and answers the question? In the discussion as it is currently written, most focus is on break points. Is this the best model tool? I am also confused by the use of AIC, and not just because there were so many models. If I look at any of the given tables in the supplement, a lower AIC score is often found for a model that was not reported, and AIC is seemingly not used for model selection. For instance, the AIC for the global ordinary least squares model (line 294) is not the lowest score on the associated table (suppl Table S2-1), it is actually for log(�DR-geo) (AIC = -25043) not for �DR (AIC = -20392). Why use AIC if you are not going to consider the best fit model? This occurred across models.

I also lost track of the methods for the result. In some cases, I don’t know that there was a method written, for example the correlations among the different metrics �-DR, �-TC and �-ClaDS at the species-level. Also, if they are all strongly correlated, why not pick one? Why continue to show all results? On lines 527-534 the caveats of the different metrics are briefly discussed – this would be more useful in the analyses as a tool for helping select a metric in the methods. If you do keep using more than one, emphasize how it effects the interpretation.

These issues with the methods need to be addressed to improve the clarity of the paper and really drive home the point without the distraction of three or four models that provide the same end. I offer other suggestions and editorial comments in the notes below.

Line 69: Define tip-level speciation rates

Line 77: Delete the space between the citation and the comma and put a period at the end of the sentence.

Line 83: Clarify here, “that limit connectivity” do you mean limit the flow of genetic material or that fragmented habitats have limited connectivity?

Line 93: freshwater habitats are more susceptible than what? Marine habitat?

Line 95: Period needed after the citation.

Line 116-121: I am confused, here, the sentences seem almost contradictory and with details that should in the methods. Maybe edit, “This study focused on species richness explanations at the assemblage and species level, capturing detailed variation among species and across space, thus helping untangle clad-specific or region-specific patterns.”

The advantage of using both the assemblage- and species-level speciation is stated as capturing more variation, but the reasoning is unclear. Please clearly indicate: What is the advantage of an assemblage-level speciation investigation? And what is the advantage of a species-level speciation investigation?

Line 122: In this paragraph, add one or two sentences clearly emphasizing the importance of this study. Why is it important to gain insight to evolutionary process of speciation, extinction, and dispersal?

Line 150: Anadromous/catadromous instead of migratory? Since there are species that migrate fully in freshwater.

Line 363: I do not understand the motivation for excluding the Cichliformes. Is there a biological or ecological reason to do so? Do they represent some type of oddity? It makes no more sense than excluding Salmoniformes. Cichliformes speciation is a good point for discussion and should continue be presented there.

Line 447: Myr ago. Also, in some places it is “Myr” and in other “myr”

Line 449: delete “higher”

Line 450: “Similarly” instead of “In the same vein”

Line 451: “genera” instead of “groups”

Line 453-454: This sentence is awkward; it reads as though the analysis in this paper examined speciation of marine fish. I suggest: “… environments only occur outside the tropics, between 27� and 40� latitude [5]. Among freshwater fishes the latitudinal position of…”

Line 460: reference?

Line 498: “find evidence of” instead of “not recover”

Line 520: define the Wallacean and Darwinian shortfalls

Figures: All figures are blurry, and quality should be enhanced.

Fig2: I cannot differentiate the lines, perhaps show below as in figure S2-5

Fig 4: write out “Next Relatedness Index” in the figure caption

Given the interest in speciation rates as related to species richness. I’m curious what this plot would look like, with species richness on the x-axis (instead of latitude) and speciation on the y-axis. Perhaps that would show an interesting pattern, maybe you’ve already considered this, and it doesn’t provide anything useful. Just a thought.

Reviewer #2: This manuscript, by Herrera-Pérez et al., presents an ambitious global-scale analysis, finding a positive relationship between absolute latitude and speciation rate in freshwater fishes, a pattern primarily driven by dynamics in the Northern Hemisphere. This challenges the long-held “tropics as cradle” hypothesis, which states that higher tropical diversity is a consequence of faster speciation rates at lower latitudes. The study is definitely in line with recent work in marine fishes and other major groups and can be a potentially high-impact contribution to the field.

The manuscript has three strengths.

1. The use of a near-complete species-level phylogeny combined with a large database of geographic ranges for over 12,000 freshwater fish species provides a good global coverage.

2. The study employs a variety of methodological tools, applying three distinct speciation rate metrics (ADR, BAMM, ClaDS) and a few statistical models (OLS, SAR, GAM, PGLS).

3. By demonstrating an inverse speciation gradient in a major freshwater vertebrate group, the work extends recent, provocative findings from marine systems to the freshwater realm, adding a crucial piece to the puzzle of what drives the global Latitudinal Diversity Gradient (LDG).

Despite these strengths, the authors can use substantial revision in three key areas to better frame the work.

1. Under-theorized Methodological Choices: The results from its diverse analytical toolkit do not sufficiently discuss the underlying assumptions and deeper implications of the chosen methods. One example of that is the well-documented methodological debate surrounding the BAMM method is not mentioned, and the reasons why different metrics (e.g., ClaDS vs. BAMM) spit out quantitatively different results (such as the location of the latitudinal breakpoint) are not explored. This leaves the analysis feeling more like a black box than a transparent and fully justified scientific investigation. The use of three different metrics to estimate speciation rates is a good step and ensures robustness. However, the discussion can be beyond simply listing that the results are robust because the metrics are highly correlated and instead engage with the reasons why these methods, which are based on quite different assumptions, might converge or diverge in their estimates (e.g., It is important to state that while the DR statistic is a computationally fast and useful proxy for speciation rate, it is also known to have high variance and can incorrectly identify rate heterogeneity even on trees that are simulated under a constant-rate process, due to the stochastic nature of branching events. Also, it has been commonly misinterpreted in the literature as a measure of net diversification rate, when it is a much better estimator of the speciation rate component. The manuscript reports a high correlation (Pearson's r>0.8) between ADR and the model-based metrics. This is an interesting finding in itself, as some studies on other taxa, such as birds, have found it to be uncorrelated with BAMM-derived speciation rates. This discrepancy warrants a brief mention in the discussion. It could be speculated that the more pronounced and discrete rate shifts in the fish phylogeny (e.g., the explosive radiations of cichlids and salmonids) create a stronger signal that is consistently picked up by both model-based and non-model-based methods, leading to higher concordance than in clades with more subtle rate variation. To provide this missing context and elevate the methodological discussion, the inclusion of a summary table is strongly recommended where the authors list the metric, core principles, key assumptions, and documented strengths and biases.

2. Problematic Interpretation of Tropical Patterns: The manuscript's narrative frames the exceptionally high speciation rates of certain tropical clades, most notably the Cichliformes, as confounding "noise" that must be statistically removed to reveal a ‘general’ underlying pattern of increasing speciation with latitude. This interpretation represents a significant conceptual weakness. These hyper-diverse clades are not an anomaly obscuring the true signal; they are a primary signal of one of the most powerful modes of diversification on the planet—explosive adaptive radiation. Treating them as outliers to be discarded rather than a distinct macroevolutionary regime to be explained and contrasted with temperate patterns weakens the study's explanatory power. I am not a fish biologist but think this might be an important point to think about.

3. Critical Omission of Biodiversity Shortfalls: The manuscript's analysis and conclusions assume that the current taxonomy of freshwater fishes is largely accurate. However, it does not address the well-documented Linnean (undescribed species) and Darwinian (unknown evolutionary relationships) shortfalls, which are particularly acute in the tropics. The high and increasing rate of discovery of cryptic species in tropical freshwater systems, especially in South America and Asia, means that many taxa represented as single, long terminal branches in the phylogeny (implying low speciation rates) may in fact be diverse, rapidly speciating complexes. This systematic underestimation of tropical speciation rates represents a major unaddressed caveat that could plausibly create an artificial inverse gradient, potentially undermining the manuscript's central conclusion.

Addressing these three points through careful revision willmake this manuscript better and I would happy to reconsider it again.

Reviewer #3: The manuscript “The latitudinal speciation gradient in freshwater fishes: higher speciation at higher

latitudes in the northern hemisphere” examines speciation rate across latitudes for the assemblage-level and species-level of freshwater fish. They used a published study on freshwater ranges and phylogenetic information to conduct their statistical analyses. They employed a range of diversification methods and regression models to investigate patterns across latitudes. By using a variety of methods they were able to evaluate a consistent pattern while acknowledging potential biases associated with each method. They were able to build a comprehensive, detailed study and give an overall robust answer. The manuscript is also well-written with objectives clearly defined. While I believe the study is for the most part well-defined, I do have a couple of minor issues that I believe the authors could respond to and not be determinental to the other research.

I was confused on the extent of the study. For the most part it seems to be a global study. While LDG is a pattern observed globally, initially, I thought this study would focus on a more specific region. They do include “across the globe” in the sentence regarding the addition of phylogenetic structure to the assemblage-level and species-level patterns. I would suggest adding that this is a global analysis either in the first or second sentence of the last paragraph of the introduction.

By using species range maps that use convex polygons with Hydrobasin layer level 8, are you overestimating species presence, and how does this affect your results? Would this affect the estimate for the latitude midpoint for each species?

Do the 111 x 111 km grid cells overestimate species presence because of the convex polygons in assemblage-level analysis

Why 111 x 111 km resolution grid? If I were to reproduce this study with another organism, I would like to know how that resolution would affect my study.

The analyses conducted at the species-level produces a very low (but significant) R2, especially compared to the assembly-level analyses. Is this a product of speciation rate or the inclusion of so many (1000+) species.

**Do you want your identity to be public for this peer review?** For information about this choice, including consent withdrawal, please see our Privacy Policy

Reviewer #1: No

Reviewer #2: No

Reviewer #3: No

---

## [Author Response · Author response to Decision Letter 1]

15 Oct 2025

Response Letter - PONE-D-25-26611

Reviewer #1:

This paper examines patterns of speciation rates of freshwater fishes across latitude, globally, finding that speciation rates increased with latitude – mainly in the northern hemisphere. The aim of the study was to describe and evaluate the latitudinal pattern of speciation in freshwater fish. And, also to determine whether speciation rates were related/responsible for the global pattern of species richness – higher richness at the equator − but this was not explicitly stated.

R: We appreciate the reviewer’s feedback and suggestions to improve our manuscript. The reviewer correctly highlights our main goals. Since we were mainly interested in describing and evaluating the latitudinal gradient of speciation in freshwater fish, which has not been previously done in the literature, we focused on this explicit goal while its relationship with the global pattern of species richness was left as part of our motivation and discussion. However, we agree with the reviewer that we should also explicitly state this secondary goal. Of course, speciation rates are ultimately related to species richness but evaluating their interdependence requires a completely different approach beyond the scope of our study. Instead, we considered such relationship between species richness and speciation to base our expectations about the latitudinal speciation gradient, following the relevant literature on the general topic of speciation gradients (Schluter & Pennell 2017, Nature) and for fishes in particular (e.g., Rabosky et al. 2018, Nature). Following another reviewer’s suggestion, we now explicitly included the relationship between species richness and speciation rates as part of our goals, and added a plot (now Figure 3) to support our discussion and interpretation.

Speciation was evaluated at the assemblage- and species-level using various metrics of tip-level speciation rate and multiple models. Findings indicated the importance of glaciations/refugia, biogeography, and evolutionary factors on speciation gradients. Where speciation was high, rates were congruent with recent radiation events of specific taxonomic groups; dispersal and environmental filters related to glaciation events likely played an important role here. This is an interesting paper, which highlights the importance of fragmentation and environmental filters on diversity.

R: We thank the reviewer for the positive appraisal of our study and the constructive comments to improve it.

The material in the introduction and discussion is well written and clear with supporting citations. My main criticism – albeit a large one – is that the methods are unruly, hard to follow, and poorly justified. I think much of this can be corrected, and as clarifications are made, I believe some of the other more minor questions I had in the discussion will be addressed.

I think the paper would benefit substantially from a flow chart that demonstrates how all of data and the models came together and how choices were/are made. There are numerous streams of information including two pattern levels, two datasets, multiple metrics of the tip-level speciation rate, several types of model structure, multiple mean calculations, and transformations. This is very complicated, and I cannot make sense of why certain things were done (e.g., the use of GAMs in addition to the other model types).

Instead of running all the models on all the metrics and showing all the results, I suggest narrowing the focus. What is the main question you are asking in this paper? What model (GAM, breakpoint, …) best fits the data and answers the question? In the discussion as it is currently written, most focus is on break points. Is this the best model tool?

R: We understand and agree with the reviewer’s concern. By trying to robustly address our main goal, we inadvertently complicated things more than necessary. In our revised version, we have carefully selected and appropriately justified our methodological choices.

First, we now provide a detailed flowchart in the main text (now Figure 1) describing how the data and models were structured based on our specific choices with their corresponding goals.

Second, given that all metrics, their different versions and transformations considered in our original manuscript showed basically the same patterns, we have reduced the number of metrics of tip-level speciation rates. More specifically, we kept the raw metrics from the three different methods (λDR, λBAMM and λClaDS) to ensure robustness of our results across distinct methods and discuss such robustness in light of recent debate regarding these methods (as suggested by other reviewers). Also, these raw metrics allow easier interpretation and comparison with previous studies relevant to ours, such as the latitudinal speciation gradient of marine fishes (Rabosky et al. 2018). In addition, for the assemblage-level analyses based on mean calculations, we also included the weighted geometric mean versions of the raw metrics, mainly because these metrics versions are less sensitive to extreme values and, more importantly, control for the disproportionate contribution of widely distributed species across different assemblages.

Third, we have narrowed our focus to our main question and goal, describe the latitudinal speciation gradient of freshwater fish speciation rates and assess whether there is a consistent relationship across the gradient or a breakpoint as suggested for marine fishes. Accordingly, we kept the breakpoint models (segmented regressions) while comparing them to OLS models to confirm the existence of a breakpoint (i.e., better model fit), followed by the SAR models fit to the relationship at each side of the breakpoint to explicitly account for spatial autocorrelation in the assemblage-level analyses. We followed the same procedure for the species-level analyses, comparing OLS vs segmented models and, after confirming that OLS models consistently provided a better fit, we fitted a PGLS model to account for phylogenetic non-independence among species.

As such, we removed the GAM models that were initially considered solely to confirm that the studied relationship was not linear, but this is already accounted for with the comparison between OLS and breakpoint models.

With these adjustments, we strongly believe that our methods are now clearly justified and easier to follow.

I am also confused by the use of AIC, and not just because there were so many models. If I look at any of the given tables in the supplement, a lower AIC score is often found for a model that was not reported, and AIC is seemingly not used for model selection. For instance, the AIC for the global ordinary least squares model (line 294) is not the lowest score on the associated table (suppl Table S2-1), it is actually for log(λDR-geo) (AIC = -25043) not for λDR (AIC = -20392). Why use AIC if you are not going to consider the best fit model? This occurred across models.

R: We apologize for the confusion and thank the reviewer for the opportunity to clarify our approach. Originally, our approach was to compare models only within the same type of speciation rate metric (e.g., OLS vs Breakpoint models for λDR) as these have the same data structure, which is a requirement for valid model comparison and selection (Burnham & Anderson, 2002). In other words, it is not appropriate to compare a model for λDR with a model for λDR-geo since they are essentially different response variables (Burnham & Anderson, 2002). To avoid confusion, we now explicitly mention this methodological aspect while also simplifying things by reducing the number of considered metrics (see previous response). In short, we used AIC-based model selection to evaluate whether the breakpoint model was a better fit than the OLS model. For the assemblage-level analyses, we further compared the fit of OLS vs SAR models to the relationships at each side of the breakpoint. For the species-level analyses, we compared the fit of OLS vs PGLS models.

I also lost track of the methods for the result. In some cases, I don’t know that there was a method written, for example the correlations among the different metrics λ-DR, λ-TC and λ-ClaDS at the species-level. Also, if they are all strongly correlated, why not pick one? Why continue to show all results? On lines 527-534 the caveats of the different metrics are briefly discussed – this would be more useful in the analyses as a tool for helping select a metric in the methods. If you do keep using more than one, emphasize how it effects the interpretation.

R: The reviewer is correct in that the correlations were not described in the methods section. We now include this methodological step. In this sense, we decided to keep the speciation rate metrics derived with different methods to ensure robustness of our results across distinct methods and discuss such robustness in light of recent debate regarding these methods. We now include more explicit discussion on this issue as suggested by other reviewers.

Still, we did reduce the number of total metrics considered, keeping only the raw metrics and their weighted geometric mean version (for the assemblage-level analyses). Please, see our response above on this issue.

These issues with the methods need to be addressed to improve the clarity of the paper and really drive home the point without the distraction of three or four models that provide the same end. I offer other suggestions and editorial comments in the notes below.

R: We thank the reviewer for providing specific guidance to improve the clarity of our study.

Line 69: Define tip-level speciation rates

R: Thanks for the suggestion. We now explicitly define tip-level speciation rates. The revised text now reads (lines 66-70):

“For example, different metrics with different assumptions have been proposed to estimate tip-level diversification rates, which can be interpreted as estimates of present-day rates of speciation or extinction of individual lineages/branches conditional on their past evolutionary history but provide more reliable estimation of speciation rates instead of net diversification [11,12].”

Line 77: Delete the space between the citation and the comma and put a period at the end of the sentence.

R: Done.

Line 83: Clarify here, “that limit connectivity” do you mean limit the flow of genetic material or that fragmented habitats have limited connectivity?

R: Thanks for pointing this out. Yes, we referred to limited spatial connectivity in fragmented habitats. We added the term ‘spatial’ to this part of the text, which also helped connect with the next paragraph in terms of how the dendritic network limits movements and thus affects processes such as dispersal and speciation.

Line 93: freshwater habitats are more susceptible than what? Marine habitat?

R: Yes. We now clarify that comparison with the marine habitat.

Line 95: Period needed after the citation.

R: Done.

Line 116-121: I am confused, here, the sentences seem almost contradictory and with details that should in the methods. Maybe edit, “This study focused on species richness explanations at the assemblage and species level, capturing detailed variation among species and across space, thus helping untangle clad-specific or region-specific patterns.”

R: We agree with the reviewer that our phrasing was confusing. In the first sentence of this part, we were actually referring to the study by Miller & Román-Palacios (2021) [reference #4] cited in the previous sentence. We now explicitly cite this study in this part to avoid the confusion. The second part of the text referred to our proposal of combining species- and assemblage-level analyses. We kept this part to explicitly state the difference between previous studies (Miller & Román-Palacios 2021) and our proposal. The revised text now reads (lines 112-116):

“Similar findings were obtained by Miller and Román-Palacios [4] for freshwater fishes when evaluating their latitudinal species richness pattern, with higher species richness in tropical basins associated to low speciation rates and low species richness in basins with high speciation rates. However, the study by Miller and Román-Palacios [4] focused on species richness explanations…”

The advantage of using both the assemblage- and species-level speciation is stated as capturing more variation, but the reasoning is unclear. Please clearly indicate: What is the advantage of an assemblage-level speciation investigation? And what is the advantage of a species-level speciation investigation?

R: Thanks for the suggestion. We now provide a clearer reasoning on using these two approaches, clearly stating their advantages. The revised text now reads (lines 119-124):

“Species-level analysis allows identifying lineage-specific patterns by considering individual species as observational units and describing trait (here, speciation rates) variation across the clade. Complementarily, assemblage-level analysis allows describing the entire geographic structure of mean trait variation by considering assemblages of species across space [29]. Thus, combining both types of approaches helps untangling clade-specific from regional-specific patterns [29].”

Line 122: In this paragraph, add one or two sentences clearly emphasizing the importance of this study. Why is it important to gain insight to evolutionary process of speciation, extinction, and dispersal?

R: Done. We appreciate this important suggestion. We added statements highlighting the importance of studying speciation rates and their geographic variation to understand biodiversity gradients, especially species richness differences among regions (Schluter & Pennell 2017, Nature), as well as describing spatial patterns of phylogenetic structure to infer the roles of the evolutionary processes that ultimately cause such biodiversity gradients. The revised text now reads (lines 128-131 and 137-142):

“To provide different perspectives on the same pattern and a more robust evaluation, we employed both assemblage-level and species-level analysis [29,30] of the geographical variation in speciation rates, which can help infer its role in explaining the observed LDG of freshwater fish species richness [5,8].”

“These patterns can provide insights into the maintenance of biodiversity gradients by describing the roles of the evolutionary process of speciation, extinction and dispersal, which are the ultimate causes underlying such gradients [8, 31]. Recent studies have evaluated these phylogenetic structure patterns in freshwater fish but focusing on specific regions independently (e.g., North America [32]; South America [33]), thus a global description is still lacking.”

Line 150: Anadromous/catadromous instead of migratory? Since there are species that migrate fully in freshwater.

R: Done.

Line 363: I do not understand the motivation for excluding the Cichliformes. Is there a biological or ecological reason to do so? Do they represent some type of oddity? It makes no more sense than excluding Salmoniformes. Cichliformes speciation is a good point for discussion and should continue be presented there.

R: Our motivation was not to neglect the importance of Cichliformes but instead consider their disproportionate effect on the evaluated pattern, which was evident when we excluded them from the whole-class analysis. In fact, we considered and discussed both patterns with and without Cichliformes in our previous version. Nevertheless, we agree with the reviewer (and the other reviewers) in that there is no a priori reason to do so and that our results should be discussed in light of their presence not its (artificial) absence from the whole-class global pattern. Accordingly, in our revised version we only kept and discussed the analysis/results considering Cichliformes and thus freshwater fishes from the whole class Actinopterygii.

Line 447: Myr ago. Also, in some places it is “Myr” and in other “myr”

Done.

Line 449: delete “higher”

Done.

Line 450: “Similarly” instead of “In the same vein”

Done.

Line 451: “genera” instead of “groups”

Done.

Line 453-454: This sentence is awkward; it reads as though the analysis in this p

---

## [Decision Letter · Decision Letter 1]

10 Nov 2025

Dear Dr. Villalobos,

Thank you for submitting your manuscript to PLOS ONE. After careful consideration, we feel that it has merit but does not fully meet PLOS ONE’s publication criteria as it currently stands. Therefore, we invite you to submit a revised version of the manuscript that addresses the points raised during the review process.

We look forward to receiving your revised manuscript.

Kind regards,

Tzen-Yuh Chiang

Academic Editor

PLOS ONE

Journal Requirements:

Reviewers' comments:

Reviewer's Responses to Questions

**Comments to the Author**

Reviewer #2: All comments have been addressed

Reviewer #3: All comments have been addressed

2. Is the manuscript technically sound, and do the data support the conclusions?

Reviewer #2: Yes

Reviewer #3: Partly

3. Has the statistical analysis been performed appropriately and rigorously?

Reviewer #2: Yes

Reviewer #3: Yes

4. Have the authors made all data underlying the findings in their manuscript fully available?

Reviewer #2: Yes

Reviewer #3: Yes

5. Is the manuscript presented in an intelligible fashion and written in standard English?

Reviewer #2: Yes

Reviewer #3: Yes

Reviewer #2: The authors have done a great job in incorporating changes based on my queries and comments on the first version of the manuscript. I am happy with thir responses and can recommend acceptance.

Reviewer #3: This is my second review of the manuscript, “The latitudinal speciation gradient in freshwater fishes: higher speciation across assemblages at higher latitudes in the northern hemisphere”. The manuscript explores speciation rate at the assemblage- and species-level of freshwater fish across a latitudinal gradient. They used three different metrics to examine speciation rates, and discussed trends with OLS and an OLS-segmented models. Moreover, they examined phylogenetic structure across regions to assess potential underlying processes influencing trends. I really enjoyed reading the manuscript. It is well written, and the authors did an exceptional amount of work. I do have some suggestions and edits to share with the authors.

Start a new paragraph at line 160: “Phylogenetic relationships were obtained from a recent species-level phylogeny of Actinopterygian fishes…”

I was happy to see that a flow chart was added. Unfortunately, the figure was very blurry, and I couldn’t make out the words. I have a feeling it's due to the upload process.

Add a “)” at the end of “R2=0.0029, p<0.001 “ in Line 327

While the authors do describe the differences between the speciation rate calculations in the discussion. I wanted the differences discussed sooner (methods) to gain a better understanding of discrepancies in species-level analyses.

New paragraph at line 332. “OLS-segmented models provided a better fit …”

I was confused why OLS results were present when OLS-segmented models were a better fit. I was making notes on the OLS results, until I came across that OLS- seg was a better fit. I suggest keeping the OLS model results in the supplemental material and focusing on the OLS-segmented results. The OLS results provide interesting information, but for the reader to stay focused, keep the results streamlined. Moreover, I would keep the AIC scores in, so the reader can see why OLS-seg was chosen.

Add a “)” to “(β=-0.001, R²=0.004, p<0.001” on line 349.

In paragraph on line 395, you discuss NRI values, but don’t reference the values or a figure. Can you add a figure reference to the paragraph?

Fig 6 caption doesn’t say what NRI represents.

The authors discuss marine fishes having a more pronounced gradient in speciation in the intro and discussion, line 440, “As expected, the global assemblage-level pattern for freshwater fishes is less pronounced than that observed for marine fishes”. Without concrete information on the difference between the two studies, this assessment feels qualitative and like a strawman argument. I doubt that it is an incorrect assessment by the authors, but as a reader, I need something tangible to compare the difference.

On line 444, “Our observed speciation peaks were found in places with high phylogenetic clustering”, I cannot fully connect your results to this statement. Are you connecting NRI and speciation values to the Neotropic (or tropical) and Nearctic regions? Later in the paragraph, you reference regions in Africa (that you found high NRI values) and neotropics (where you found sig. NRI values), but you are connecting results from two sections, making it slightly hard to follow. The reader might need some hand-holding or references to figures.

**Do you want your identity to be public for this peer review?** For information about this choice, including consent withdrawal, please see our Privacy Policy

Reviewer #2: No

Reviewer #3: No

---

## [Author Response · Author response to Decision Letter 2]

11 Nov 2025

Response Letter - PONE-D-25-26611.R1

Review Comments to the Author

Reviewer #2:

The authors have done a great job in incorporating changes based on my queries and comments on the first version of the manuscript. I am happy with their responses and can recommend acceptance.

R: We appreciate the constructive and positive comments of the reviewer, which certainly helped to improve our manuscript. We are glad that our revised version met their standards and can be considered acceptable for publication.

Reviewer #3:

This is my second review of the manuscript, “The latitudinal speciation gradient in freshwater fishes: higher speciation across assemblages at higher latitudes in the northern hemisphere”. The manuscript explores speciation rate at the assemblage- and species-level of freshwater fish across a latitudinal gradient. They used three different metrics to examine speciation rates, and discussed trends with OLS and an OLS-segmented models. Moreover, they examined phylogenetic structure across regions to assess potential underlying processes influencing trends. I really enjoyed reading the manuscript. It is well written, and the authors did an exceptional amount of work.

R: We are grateful to the reviewer for their encouraging comments and thoughtful criticisms. We truly believe that such careful reviews have strengthen our study.

I do have some suggestions and edits to share with the authors.

Start a new paragraph at line 160: “Phylogenetic relationships were obtained from a recent species-level phylogeny of Actinopterygian fishes…

R: Done.

I was happy to see that a flow chart was added. Unfortunately, the figure was very blurry, and I couldn’t make out the words. I have a feeling it's due to the upload process.

Add a “)” at the end of “R2=0.0029, p<0.001 “ in Line 327

R: Done.

Regarding the figure, yes, it must be the upload process. The figure can be accessed by clicking on the top right corner of the corresponding page where the figure was uploaded. There, there is a link to download the figure embedded in each corresponding page. Anyhow, to facilitate visualization, we are pasting a version of this flowchart figure below:

While the authors do describe the differences between the speciation rate calculations in the discussion. I wanted the differences discussed sooner (methods) to gain a better understanding of discrepancies in species-level analyses.

R: Thanks for the observation. We agree with the reviewer and have now added the following sentences to the corresponding paragraph in the methods section (“Speciation rates”; lines 203-209):

“Each of these speciation rate metrics have important differences, with BAMM identifying discrete shifts in diversification regimes (i.e., cohorts) that tend to reduce rate heterogeneity along the phylogeny compared to DR and ClaDS. Conversely, DR and ClaDS recover higher rate heterogeneity compared to BAMM, with ClaDS being able to identify many small and frequent shifts that result in more heterogeneous rates among lineages [12]. Accordingly, considering these three metrics allowed us to evaluate the robustness of our studied patterns beyond methodological differences.”

New paragraph at line 332. “OLS-segmented models provided a better fit …”

R: Done.

I was confused why OLS results were present when OLS-segmented models were a better fit. I was making notes on the OLS results, until I came across that OLS- seg was a better fit. I suggest keeping the OLS model results in the supplemental material and focusing on the OLS-segmented results. The OLS results provide interesting information, but for the reader to stay focused, keep the results streamlined. Moreover, I would keep the AIC scores in, so the reader can see why OLS-seg was chosen.

R: We understand the reviewer’s concern. Certainly, we do want the readers to stay focused, but we also want them to have the elements for considering our study within the broader literature, mainly the one related to the other main group of fishes, the marine fishes. Accordingly, we considered it important to keep the presentation of the OLS results to describe the general/global relationship between latitude and speciation rates, which is the most common in the literature being used to describe the pattern for marine fishes and other taxa (e.g. mammals, birds, amphibians). Also, we think that showing the OLS and OLS-segmented results helps to explicitly/clearly illustrate the fact that the relationship is indeed more complex than a simple linear relationship, which in turn allows a more comprehensive discussion of our findings with those of others.

Add a “)” to “(β=-0.001, R²=0.004, p<0.001)” on line 349.

R: Done.

In paragraph on line 395, you discuss NRI values, but don’t reference the values or a figure. Can you add a figure reference to the paragraph?

R: Done. We have included a reference to Figure 6 that shows those patterns.

Fig 6 caption doesn’t say what NRI represents.

R: Done.

The authors discuss marine fishes having a more pronounced gradient in speciation in the intro and discussion, line 440, “As expected, the global assemblage-level pattern for freshwater fishes is less pronounced than that observed for marine fishes”. Without concrete information on the difference between the two studies, this assessment feels qualitative and like a strawman argument. I doubt that it is an incorrect assessment by the authors, but as a reader, I need something tangible to compare the difference.

R: We appreciate the reviewer’s comment and opportunity to clarify our message. We agree with the reviewer in that we need something more tangible to compare our results with those of marine fishes. Therefore, in our new revised version, we have expanded this part of the discussion to clearly relate our findings to those of marine fishes, thus providing more tangible evidence to conduct such comparison, including reference to our figures.

In particular, we added the following text to the cited paragraph (lines 445-458):

“The latitudinal speciation gradient of freshwater fishes was less consistent than that observed for marine fishes. In marine fishes, Rabosky et al. [5] found a consistently positive global relationship between latitude and speciation, regardless of their considered metrics (BAMM or DR). In contrast, the global relationship for freshwater fishes was not consistent between these two metrics, with BAMM showing the same pattern to marine fishes whereas DR showed no significant latitudinal trend. Indeed, marine fishes exhibited a clear u-shaped pattern between latitude and speciation, with high speciation rates at both northern and southern latitudes [5]. Instead, freshwater fishes did not show such a clear pattern but a more heterogenous one with higher variation of speciation rates in tropical regions (Fig. 2). This difference between marine and freshwater fishes is mainly given by tropical freshwater assemblages exhibiting very high speciation rates, whereas tropical marine assemblages showed consistently low speciation rates [5]. Our observed speciation peaks in the tropics were located in places with high phylogenetic clustering, indicating the co-occurrence of recently derived and/or closely related lineages within the assemblages (Fig 6), which is consistent with recent findings for freshwater fishes at the basin level [4], where the highest speciation rates are located in recently colonized basins.”

On line 444, “Our observed speciation peaks were found in places with high phylogenetic clustering”, I cannot fully connect your results to this statement. Are you connecting NRI and speciation values to the Neotropic (or tropical) and Nearctic regions? Later in the paragraph, you reference regions in Africa (that you found high NRI values) and neotropics (where you found sig. NRI values), but you are connecting results from two sections, making it slightly hard to follow. The reader might need some hand-holding or references to figures.

R: Thanks for the important suggestion. In following it, we have added an explicit mention to the “tropics” and a reference to the relevant figure for this pattern (Fig 6).

---

## [Editor Report · Decision Letter 2]

1 Dec 2025

The latitudinal speciation gradient in freshwater fishes: higher speciation across assemblages at higher latitudes in the northern hemisphere

PONE-D-25-26611R2

Dear Dr. Fabricio Villalobos,

We’re pleased to inform you that your manuscript has been judged scientifically suitable for publication and will be formally accepted for publication once it meets all outstanding technical requirements.

Kind regards,

Tzen-Yuh Chiang

Academic Editor

PLOS ONE
---

## [Editor Report · Acceptance letter]

PONE-D-25-26611R2

PLOS One

Dear Dr. Villalobos,

I'm pleased to inform you that your manuscript has been deemed suitable for publication in PLOS One. Congratulations! Your manuscript is now being handed over to our production team.

Kind regards,

on behalf of

Dr. Tzen-Yuh Chiang

Academic Editor

PLOS One